# Optimal Dynamic Regret in LQR Control

**Dheeraj Baby**
Department of Computer Science
UC Santa Barbara
dheeraj@ucsb.edu

**Yu-Xiang Wang**
Department of Computer Science
UC Santa Barbara
yuxiangw@cs.ucsb.edu

## Abstract

We consider the problem of nonstochastic control with a sequence of quadratic losses, i.e., LQR control. We provide an efficient online algorithm that achieves an optimal dynamic (policy) regret of $\tilde{O}(\max\{n^{1/3}\mathcal{TV}(M_{1:n})^{2/3}, 1\})$, where $\mathcal{TV}(M_{1:n})$ is the total variation of any oracle sequence of *Disturbance Action* policies parameterized by $M_1, ..., M_n$ — chosen in hindsight to cater to unknown nonstationarity. The rate improves the best known rate of $\tilde{O}(\sqrt{n(\mathcal{TV}(M_{1:n}) + 1)})$ for general convex losses [Zhao et al., 2022] and we prove that it is information-theoretically optimal for LQR. Main technical components include the reduction of LQR to online linear regression with delayed feedback due to Foster and Simchowitz [2020], as well as a new *proper* learning algorithm with an optimal $\tilde{O}(n^{1/3})$ dynamic regret on a family of "minibatched" quadratic losses, which could be of independent interest.

## 1 Introduction

This paper studies the linear quadratic regulator (LQR) control problem which is a specific instantiation of the more general RL framework where the evolution of states follows a predefined linear dynamics. At each round $t \in [n] := \{1, \ldots, n\}$, the agent is at state $x_t \in \mathbb{R}^{d_x}$. Based on the state, the agent select a control input $u_t \in \mathbb{R}^{d_u}$. The next state evolves according to the law:

$$x_{t+1} = Ax_t + Bu_t + w_t,$$

where $A$ and $B$ are system matrices known to the agent. $w_t \in \mathbb{R}^{d_x}$ is a disturbance term that can be selected by a potentially adaptive adversary. We assume that $\|w_t\|_2 \leq 1$. This disturbance term reflects the perturbation from the ideal linear state transition arising due to environmental factors that could be difficult to model. The loss suffered by playing the control $u$ at state $x$ is given by $\ell(x, u) := x^T R_x x + u^T R_u u$, where $R_x, R_u \succcurlyeq 0$, that are apriori fixed and known.

Recently there has been a surge of interest in viewing this classical LQR problem under the lens of online learning [Hazan, 2016]. The work of Agarwal et al. [2019] places regret of the agent against a set of benchmark policies as the central notion to evaluate learner's performance. Following Agarwal et al. [2019], Foster and Simchowitz [2020] we adopt the class of disturbance action policies (DAP) as our benchmark class:

**Definition 1.** *(Disturbance action policies, [Foster and Simchowitz, 2020]). Let $M = (M^{[i]})_{i=1}^m$ denote a sequence of matrices $M^{[i]} \in \mathbb{R}^{d_u \times d_x}$. We define the corresponding disturbance action policies (DAP) $\pi^M$ as:*

$$\pi_t^M(x_t) = -K_\infty x_t - q^M(w_{1:t-1}), \tag{1}$$

*where $q^M(w_{1:t-1}) = \sum_{i=1}^m M^{[i]} w_{t-i}$ and $K_\infty$ as in Eq.(4). We are interested in DAPs for which the sequence $M$ belongs to the set:*

$$\mathcal{M}(m, R, \gamma) := \{M = (M^{[i]})_{i=1}^m : \|M^{[i]}\|_{op} \leq R\gamma^{i-1}\}, \tag{2}$$

*where $m$, $R$ and $\gamma$ are algorithm parameters.*

This class is known to be sufficiently rich to approximate many linear controllers. A policy takes in the past history and current state as input and produces a control signal as output. Let's denote $M_{1:n} := (M_1, \ldots, M_n)$ to be a sequence of DAP policies such that at time $t$, the control signal is selected using the policy parameterized by $M_t$ (see Eq.(1)). We denote $x_t^{M_{1:n}}$ to be the state reached at round $t$ by playing the sequence of policies defined by parameters $M_{1:t-1}$ in the past. Similarly $u_t^{M_{1:n}}$ is used to denote the control signal produced by the policy $M_t$. The *universal dynamic regret* of the learner against the policy sequence $M_{1:n}$ is defined as:

$$R(M_{1:n}) = \sum_{t=1}^n \ell(x_t^{\text{alg}}, u_t^{\text{alg}}) - \ell(x_t^{M_{1:n}}, u_t^{M_{1:n}}), \tag{3}$$

where $(x_t^{\text{alg}}, u_t^{\text{alg}})$ denotes the state and control signal of the learner at round $t$. Note that the policy sequence $M_{1:n}$ can be *any* valid sequence of DAP polices. The main focus of this paper is to design algorithms that can control the dynamic regret against a sequence of reference policies as a function of the time horizon $n$ and the a path variation of the DAP parameters of the comparator $M_{1:n}$. We remark that the comparator polices $M_{1:n}$ can be chosen in hindsight and potentially unknown to the learner.

Whenever $M_{1:n} = (M, \ldots, M)$ for a fixed parameter $M$, we recover the notion of static regret. However the notion of static regret is not befitting for non-stationary environments. For best performance under a non-stationary environment, a controller may have to choose different policies at different time steps that can counter-act the disturbances and guide the dynamics properly. Hence, we aim to control the dynamic regret which allows us to be competent against a sequence of potentially time-varying polices chosen in hindsight. We remark that our algorithm automatically adapts to the level of non-stationarity in the hindsight sequence of policies.

Next, we take a digression and discuss a desirable property for the design of algorithms for LQR control.

**Proper learning in LQR control.** Proper learning is an online learning paradigm where the decisions of the learner are required to obey some user specified safety constraints. On the other hand, improper learning framework allows the learner to disregard such safety constraints. The paradigm of improper learning may not be attractive in certain applications where safety is a paramount concern. Improper algorithms can possibly take the system through trajectories that are deemed to be risky. It is desirable to avoid such behaviours in physical systems such as self driving cars, control of medical ventilators, robotic control [Levine et al., 2016] and cooling data centers [Cohen et al., 2018]. When translated into the LQR control problem, we regard the benchmark pool $\mathcal{M}(m, R, \gamma)$ defined in Eq.(2) as a space of safe policies. So to avoid risky behaviours, at any round, the learner plays a control signal that is recommended by a DAP policy in the class $\mathcal{M}(m, R, \gamma)$.

Below are our contributions:

- We develop an optimal universal dynamic regret minimization algorithm for the general mini-batch linear regression problem (see Theorem 5).
- Applying the reduction of Foster and Simchowitz [2020] from LQR problem to online linear regression, the above result lends itself to an algorithm for controlling the dynamic regret of the LQR problem (Eq.(3)) to be $\tilde{O}^*(n^{1/3}[\mathcal{TV}(M_{1:n})]^{2/3})$, where $\mathcal{TV}$ denotes the total variation incurred by the sequence of DAP policy parameters in hindsight (see Corollary 10). $O^*$ hides the dependencies in dimensions and system parameters.
- We show that the aforementioned dynamic regret guarantee is minimax optimal modulo dimensions and factors of $\log n$ (see Theorem 11).
- The resulting algorithm is also strongly adaptive, in the sense that the static regret against a DAP policy in any local time window is $O^*(\log n)$.

**Notes on novelty and impact.** As discussed before, the reduction of Foster and Simchowitz [2020] casts LQR problem to an instance of proper online linear regression. In the context of regression, proper learning means that the decisions of the learner belongs to a user specified convex domain. The main challenge in developing aforementioned contributions rests on the design of an optimal

universal dynamic regret minimization algorithm for online linear regression under the setting of *proper learning*. We are not aware of any such algorithms in the literature to-date and the problem remains open. However, there exists an improper algorithm from Baby and Wang [2021] for controlling the desired dynamic regret. Given this fact, the design of our algorithm is facilitated by coming up with *new* black-box reductions (see Section 4) that can convert an improper algorithm for non-stationary online linear regression to a proper one. There are improper to proper black-box reduction schemes given in the influential work of Cutkosky and Orabona [2018]. However, they are developed to support general convex or strongly convex (see Definition 4) losses. The linear regression losses arising in our setting are exp-concave (see Definition 3) which enjoy strong curvature only in the direction of the gradients as opposed to uniformly curved strongly convex losses. Hence the reduction scheme of Cutkosky and Orabona [2018] is inadequate to provide fast regret rates in our setting. In contrast, we develop novel reduction schemes that carefully take the non-uniform curvature of the linear regression losses into account so as to facilitate fast dynamic regret rates (see Section 4.2). The construction of this new reduction scheme requires non-trivial adaptation of the ideas in Cutkosky and Orabona [2018]. We remark that the algorithm ProDR.control developed in Section 4 can be impactful in general online learning literature. That the non-stationary LQR problem can be *optimally* solved using ProDR.control is a testament to this fact. Further our algorithm is out-of-the-box applicable to more general settings such as non-stationary multi-task linear regression, which is beyond the current scope. The lower bound we provide in Theorem 11 is also applicable to the more general problem of online non-parametric regression against a Besov space / class of Total Variation bounded functions [Rakhlin and Sridharan, 2014] (see Section 5 for more details). The main contribution here is that we provide a new lower bounding strategy that characterizes the correct rate wrt both $n$ and the radius (or path-variation) of the non-parametric function class. This is in contrast with Rakhlin and Sridharan [2014] who establish the correct dependency only wrt $n$. Attaining the correct dependencies wrt both $n$ and the radius / path-variation is imperative in implying a dynamic regret lower bound for the LQR problem.

The rest of the paper is organized as follows. In Section 2, we cover the necessary preliminaries on LQR control. Section 3 discusses relevant literature. In Section 4, we develop a proper algorithm for non-stationary online linear regression. In Section 5, we apply the results of Section 4 to provide an algorithm for non-sationary LQR control and prove its minimax optimality. This is followed by conclusion and open problems in Section 6. We provide a concise overview of the results from Baby and Wang [2022] in Appendix A which we build upon. All proofs are given in Appendix B.

## 2    Preliminaries

We start with a brief overview of the LQR problem for the sake of completeness. The material of this section closely follows Foster and Simchowitz [2020]. The definitions and notations introduced in this section will be used throughout the paper.

A linear control law is given by $u_t = -Kx_t$ for a controller $K \in \mathbb{R}^{d_u \times d_x}$. A linear controller $K$ is said to be stabilizing if $\rho(A - BK) < 1$ where $\rho(A - BK)$ is the maximum of the absolute values of the eigenvalues of $A - BK$. We assume that there exists a stabilizing controller for the system $(A, B)$. For such systems, there exists a unique matrix $P_\infty$ which is the solution to the equation:

$$P = A^T P A + R_x - A^T P B (R_u + B^T P B)^{-1} B^T P A.$$

The solution $P_\infty$ is called the infinite horizon Lyapunov matrix. It is an intrinsic property of the system $(A, B)$ and characterizes the optimal infinite horizon cost for control in the absence of noise [Bertsekas, 2005]. We also define the optimal state feedback controller

$$K_\infty := (R_u + B^T P_\infty B)^{-1} B^T P_\infty A, \tag{4}$$

the steady state covariance matrix:

$$\Sigma_\infty := R_u + B^T P_\infty B,$$

and the closed loop dynamics matrix: $A_{\text{cl},\infty} := A - BK_\infty$.

Foster and Simchowitz [2020] shows that the problem of controlling the regret in the LQR problem can be reduced to online linear regression problem with delays. Specifically we have the following fundamental result due to Foster and Simchowitz [2020].

**Proposition 2.** *Suppose the learner plays policy of the form $\pi_t^{alg}(x) = -K_\infty x + q^{M_t^{alg}}(w_{1:t-1})$. Let the comparator policies take the form $\pi_t(x) = -K_\infty x + q^{M_t}(w_{1:t-1})$ for a sequence of matrices $M_{1:n}$ chosen in hindsight. Then the dynamic regret against the policies $\pi := (\pi_1, \ldots, \pi_n)$ satisfies:*

$$R_n(\pi) \leq O(1) + \sum_{t=1}^{n} \hat{A}_t(M_t^{alg}, w_{t:t+h}) - \hat{A}_t(M_t, w_{t:t+h}),$$

*where the parameters involved in the inequality are defined as below:* $\hat{A}_t(M, w_{t:t+h}) := \|q^M(w_{1:t-1}) - q_{\infty;h}(w_{t:t+h})\|_{\Sigma_\infty}^2$. $q_{\infty;h}(w_{t:h+t}) := \sum_{i=t+1}^{t+h} \Sigma_\infty^{-1} B^T (A_{cl,\infty})^{i-1-t} P_\infty w_i$. $h := 2(1-\gamma_\infty)^{-1} \log(\kappa_\infty^2 \beta_*^2 \Psi_* \Gamma_*^2 n^2)$. $\gamma_\infty := \|I - P + \infty^{-1/2} R_x P_\infty^{1/2}\|_{op}^{1/2}$. $\kappa_\infty := \|P_\infty^{1/2}\|_{op} \|P_\infty^{-1/2}\|_{op}$. $\beta_* := \max\{1, \lambda_{min}^{-1}(R_u), \lambda_{min}^{-1}(r_x)\}$. $\Psi_* = \max\{1, \|A\|_{op}, \|B\|_{op}, \|R_x\|_{op}, \|R_u\|_{op}\}$. $\Gamma_* := \max\{1, \|P_\infty\|_{op}\}$

Observe that the losses $\hat{A}_t(M, w_{t:t+h}) := \|q^M(w_{1:t-1}) - q_{\infty;h}(w_{t:t+h})\|_{\Sigma_\infty}^2 = \hat{A}_t(M, w_{t:t+h}) := \|\Sigma_\infty^{1/2} q^M(w_{1:t-1}) - \Sigma_\infty^{1/2} q_{\infty;h}(w_{t:t+h})\|_2^2$ are essentially linear regression losses. The quantity $\Sigma_\infty^{1/2} q^M(w_{1:t-1})$ is a linear map from the matrix sequence $M$ to $\mathbb{R}^{d_u}$. However, there is one caveat in that the bias vector at round $t$ given by $\Sigma_\infty^{1/2} q_{\infty;h}(w_{t:t+h})$ is only available at round $t + h = t + O(\log n)$. This issue of delayed feedback can be directly handled using the delayed to non-delayed online learning reduction from Joulani et al. [2013].

## 3 Related work

In this section, we review recent progress at the intersection of control and online convex optimization (OCO) that are most relevant to our work.

**Online control**. The idea of using tools from OCO for general control problem was proposed in Agarwal et al. [2019]. They place the notion of regret against the class of DAP policies as the central performance measure. The DAP class is also shown to be sufficiently rich to approximate a wide class of linear state-feedback controllers. Under general convex losses, they propose a reduction to OCO with memory [Merhav et al., 2000, Anava et al., 2015] and derives $O(\sqrt{n})$ regret when the system matrices $(A, B)$ are known. For the case of unknown system, Hazan et al. [2020] provides $O(n^{2/3})$ regret via system identification techniques. When the losses are strongly convex and sub-quadratic, Simchowitz [2020] strengthens these results to attain $\tilde{O}(n)$ regret for known systems and $\tilde{O}(\sqrt{n})$ when the system is unknown. For partially observable systems strong regret guarantees are provided in Simchowitz et al. [2020]. Luo et al. [2022] provides an $O(n^{3/5})$ dynamic regret bound for the case when the system matrices $(A_t, B_t)$ can change over time. Their results are incompatible to ours in that they consider unknown dynamics, stochastic disturbances and the dynamic regret compete with controllers that are pointwise optimal (restricted dynamic regret), while we assume known dynamics, adversarial disturbances and compete with an arbitrary sequence of controllers (i.e., universal dynamic regret).

Next, we clarify the differences from other existing work on the nonstochastic control problems [Gradu et al., 2020a,b, Cassel and Koren, 2020, Zhang et al., 2021b, Shi et al., 2020, Goel and Hassibi, 2021, Zhao et al., 2022].

Gradu et al. [2020a], Cassel and Koren [2020] studied online control in the partially observed cases with bandit feedback, but did not consider the problem of non-stationarity with dynamic regret. Gradu et al. [2020b], Zhang et al. [2021b] studied the adaptive regret in nonstochastic control problems, which is an alternative metric to capture the performance of the learning controller in non-stationary environments. Our algorithm uses a reduction to adaptive regret too, but it is highly nontrivial to show that one can tweak adaptive regret minimizing algorithms into ones that achieve optimal dynamic regret. Moreover, our algorithm is the first that achieves logarithmic adaptive regret for nonstochastic LQR control problems too. In contrast, Gradu et al. [2020b], Zhang et al. [2021b] focused on the slower adaptive regret in the general convex loss cases.

To the best of our knowledge, Goel and Hassibi [2021] and Zhao et al. [2022] are the only existing work that considered dynamic regret in non-stochastic control.

Goel and Hassibi [2021] used tools from $H_\infty$ control and derived a controller with exact minimax optimal dynamic regret against an oracle controller that sees the whole sequence of disturbances

and chooses an optimal sequence of control actions. But the optimal dynamic regret against the sequence of control actions given by the unrealizable oracle controller is linear in $n$ in general (see an explicit lower bound from Goel and Hassibi [2020]). It is unclear whether this oracle controller can be realized by a sequence of time-varying DAP controllers. If so, then our results would imply regret bound against the optimal sequence of control actions too. Comparing to the exact minimax regret of the $H_\infty$ style controller, our regret bound would *adapt to each problem instance*, and is sublinear whenever the approximating sequence of DAP controllers has sublinear total variation.

Zhao et al. [2022] studied the universal dynamic (policy) regret problem similar to ours, but works for a broader family convex loss functions. Their regret bound $O(\sqrt{n(1 + C_n)})$ is optimal for the convex loss family. Our results show that the optimal regret improves to $\tilde{O}(n^{1/3}C_n^{2/3} \vee 1)$ when specializing to the LQR problem where the losses are quadratic. On the technical level, Zhao et al. [2022] used a reduction to the dynamic regret of OCO with memory, while we reduced to the dynamic regret of OCO with delayed feedback.

**Dynamic regret minimization in online learning**. There is a rich body of literature on dynamic regret (Eq.(5)) minimization. As discussed in Section 2, the non-staionary LQR problem can be reduced to an instance of linear regression losses which are exp-concave on compact domains. There is a recent line of research [Baby and Wang, 2021, 2022] that provides optimal universal dynamic regret rates under exp-concave losses. However, the algorithm of Baby and Wang [2021] is improper, in the sense that the iterates of the learner can lie outside the feasibility set. The work of Baby and Wang [2022] ameliorates this issue to some extend by providing proper algorithms for the particular case of $L_\infty$ constrained (box) decisions sets. The DAP policy space in Definition 1 is indeed not an $L_\infty$ ball. We note that if improper learning is allowed in the LQR problem, one can run the algorithms of Baby and Wang [2021, 2022] to attain optimal dynamic regret rates. The proper learning algorithms such as Zinkevich [2003], Zhang et al. [2018a], Cutkosky [2020], Jacobsen and Cutkosky [2022] control dynamic regret for general convex losses. However, they are not adequate to optimally minimize dynamic regret under curved losses that are strongly convex or exp-concave. The notion of restrictive dynamic regret introduced in Besbes et al. [2015] competes with a sequence of minimizers of the losses. This notion of regret can sometimes be overly pessimistic as noted in Zhang et al. [2018a]. There is a series of work in the direction of dynamic regret minimization in OCO such as Jadbabaie et al. [2015], Yang et al. [2016], Mokhtari et al. [2016], Chen et al. [2018], Zhang et al. [2018b], Goel and Wierman [2019], Baby and Wang [2019], Zhao et al. [2020], Zhao and Zhang [2021], Zhao et al. [2022], Chang and Shahrampour [2021], Baby and Wang [2020], Baby et al. [2021b], Chatterjee and Goswami [2022], Baby et al. [2021a], Raj et al. [2020]. However, to the best of our knowledge none of these works are known to attain the optimal universal dynamic regret rate for the setting of online linear regression.

**Dynamic regret for OCO vs Dynamic (Policy) regret for Control.** We emphasize that the regret in Eq.(3) is dynamic *policy* regret [Anava et al., 2015, Zhao et al., 2022]. The states visited by the reference policy is counterfactual and is different from that of the learner's trajectory which we observe. This is very different from the standard OCO framework where the state of both the learner and adversary are same. So bounding the policy regret seems qualitatively harder than bounding the regret in an OCO setting. Nevertheless, for the LQR problem, the fact that there exists a reduction [Foster and Simchowitz, 2020] from the problem of controlling policy regret to the problem of controlling the standard OCO regret is remarkable.

**Strongly adaptive regret minimization.** There is also a complementary body of literature on strongly adaptive algorithms that focus on controlling the static regret in any local time window. For example, the algorithm of Daniely et al. [2015], Jun et al. [2017] can lead to $\tilde{O}(\sqrt{|I|})$ static regret in any interval of $I \subseteq [n]$ under convex losses. When the losses are exp-concave the algorithm of Hazan and Seshadhri [2007], Adamskiy et al. [2016], Zhang et al. [2021a] can lead to $O(\log n)$ static regret in any interval.

# 4 Non-stationary "mini-batch" Linear Regression

In view of Proposition 2, the losses of interest are linear regression type losses. So we take a digression in this section and study the problem of controlling dynamic regret in a general linear regression setting.

## 4.1 Linear regression framework

Consider the following linear regression protocol.

- At round $t$, nature reveals a co-variate matrix $A_t \in \mathbb{R}^{p \times d}$.
- Learner plays $z_t \in \mathcal{D} \subset \mathbb{R}^d$.
- Nature reveals the loss $f_t(z) = \|A_t z - b_t\|_2^2$.

Under the above regression framework, we are interested in controlling the universal dynamic regret against an arbitrary sequence of predictors $u_1, \ldots, u_n \in \mathcal{D}$ (abbreviated as $u_{1:n}$) :

$$R_n(u_{1:n}) = \sum_{t=1}^{n} f_t(z_t) - f_t(u_t). \tag{5}$$

Dynamic regret is usually expressed as a function of $n$ and a path variational that captures the smoothness of the comparator sequence. We will focus on the path variational defined by:

$$\mathcal{TV}(u_{1:n}) = \sum_{t=2}^{n} \|u_t - u_{t-1}\|_1.$$

Below are the list of assumptions made:

**Assumption 1**. Let $a_{t,i} \in \mathbb{R}^d$ be the $i^{th}$ row vector of $A_t$. We assume that $\|a_{t,i}\|_1 \le \alpha$ for all $t \in [n]$ and $i \in [p]$. Further $\|b_t\|_1 \le \sigma$ for all $t$.

**Assumption 2**. For any $x \in \mathcal{D}$, $\|x\|_1 \le \chi$ and $\|x\|_\infty \le \tilde{R}$.

We refer this setting as mini-batch linear regression since the loss at round $t$ can be written as a sum of a batch of quadratic losses: $f_t(z) = \sum_{i=1}^{p} \left(z^T a_{t,i} - b_t[i]\right)^2$.

**Terminology**. For a convex loss function $f$, we abuse the notation and take $\nabla f(x)$ to be a sub-gradient of $f$ at $x$. We denote $\mathcal{D}_\infty(\tilde{R}) := \{x \in \mathbb{R}^d : \|x\|_\infty \le \tilde{R}\}$.

Linear regression losses belong to a broad family of convex loss functions called exp-concave losses:

**Definition 3.** *A convex function $f$ is $\alpha$ exp-concave in a domain $\mathcal{D}$ if for all $x, y \in \mathcal{D}$ we have $f(y) \ge f(x) + \nabla f(x)^T(x - y) + \frac{\alpha}{2}(\nabla f(x)^T(x - y))^2$.*

We refer the reaeder to Cesa-Bianchi and Lugosi [2006] and Hazan et al. [2007] for more detailed expositions on exp-concavity.

The losses $f_t(z) = \|A_t z - b_t\|_2^2$ are $(2R)^{-1}$ exp-concave if $f(z) \le R$ for all $z \in \mathcal{D}$ (see Lemma 2.3 in Foster and Simchowitz [2020]).

**Definition 4.** *A convex function $f$ is $\sigma$ strongly convex wrt $\|\cdot\|_2$ norm in a domain $\mathcal{D}$ if for all $x, y \in \mathcal{D}$ we have $f(y) \ge f(x) + \nabla f(x)^T(x - y) + \frac{\sigma}{2}\|x - y\|_2^2$.*

We note that if the matrix $A_t$ is rank deficient, then the losses $f_t(z)$ cannot be strongly convex. Moving forward we do not impose any restrictive assumptions on the rank of $A_t$. As mentioned in Remark 12, the covariate matrix that arise in the reduction of the LQR problem to linear regression is not in general full rank. So we target a solution that can handle general covariate matrices irrespective of their rank.

## 4.2 The Algorithm

Starting point of our algorithm design is the work of Baby and Wang [2022]. They provide an algorithm that attains optimal dynamic regret when the losses are exp-concave. However, their setting works only in a very restrictive setup where the decision set is an $L_\infty$ constrained box. Consequently, we cannot directly apply their results to the linear regression problem of Section 4 whenever the decision set $\mathcal{D}$ is a general convex set.

An online learner is termed proper if the decisions of the learner are guaranteed to lie within the feasibility set $\mathcal{D}$. Otherwise it is called improper. A recent seminal work of Cutkosky and Orabona [2018] proposes neat reductions that can convert an improper online learner to a proper one, whenever

> ProDR.control: Inputs - Decision set $\mathcal{D}$, $G > 0$, a surrogate algorithm $\mathcal{A}$ which ensures low dynamic regret under general exp-concave losses against any comparator sequence in some $\mathcal{D}' \supset \mathcal{D}$. Here $\mathcal{D}'$ is a compact and convex set. Note that such an algorithm $\mathcal{A}$ may produce iterates outside $\mathcal{D}$. (See Theorem 5 for a specific choice of $\mathcal{A}$.)
>
> 1. At round $t$, receive $w_t$ from $\mathcal{A}$.
> 2. Receive co-variate matrix $A_t := [a_{t,1}, \ldots, a_{t,p}]^T$.
> 3. Play $\hat{w}_t \in \operatorname{argmin}_{x \in \mathcal{D}} \max_{i=1,\ldots,p} |a_{t,i}^T(x - w_t)|$.
> 4. Let $\ell_t(w) = f_t(w) + G \cdot S_t(w)$, where $f_t(w) = \|A_t w - b_t\|_2^2$ and $S_t(w) = \min_{x \in \mathcal{D}} \max_{i=1,\ldots,p} |a_{t,i}^T(x - w)|$.
> 5. Send $\ell_t(w)$ to $\mathcal{A}$.

Figure 1: ProDR.control: An algorithm for non-stationary and proper linear regression.

the losses are convex. Following this line of research, we can aim to convert the algorithm of Baby and Wang [2022] that works exclusively on box decision set to one that can support arbitrary convex decision sets by coming up with suitable reduction schemes. However, the specific reduction scheme proposed in Cutkosky and Orabona [2018] is inadequate to yield fast dynamic rates for exp-concave losses. Our algorithm ProDR.control (Fig.1, **Pro**per **D**ynamic **R**egret.control) is a by-product of constructing new reduction schemes to circumvent the aforementioned problem for the case of linear regression losses. We expand upon these details below.

In ProDR.control, we maintain a surrogate algorithm $\mathcal{A}$, which is chosen to be the algorithm of Baby and Wang [2022] that produces iterates $w_t$ in an $L_\infty$ norm ball (box), $\mathcal{D}_\infty$, that encloses the actual decision set $\mathcal{D}$. We take $\mathcal{D}' = \mathcal{D}_\infty$ as per the notations in Fig.1. Since $w_t$ can be infeasible, we play $\hat{w}_t$ obtained via a special type of projection of $w_t$ onto $\mathcal{D}$ which is formulated as a min-max problem in Line 3 of Fig.1. In Line 4, we construct surrogate losses $\ell_t$ to be passed to the algorithm $\mathcal{A}$. The surrogate loss penalises $\mathcal{A}$ for making predictions outside $\mathcal{D}$. We will show (see Lemma 15 in Appendix) that the instantaneous regret satisfies $f_t(\hat{w}_t) - f_t(u_t) \le \ell_t(w_t) - \ell_t(u_t)$, where $u_t \in \mathcal{D}$ is the comparator at round $t$. Thus the dynamic regret of the proper iterates $\hat{w}_t$ wrt linear regression losses is upper bounded by the dynamic regret of the surrogate algorithm $\mathcal{A}$ on the losses $\ell_t$ and box decision set.

The design of the min-max barrier $S_t(w)$ is driven to ensure exp-concavity of the surrogate losses $\ell_t(w) = f_t(w) + G \cdot S_t(w)$. We capture its intuition as follows. We start by observing that since $\nabla^2 f_t(w) = 2 A_t^T A_t$, the linear regression losses $f_t$ exhibits strong curvature along the row-space of $A_t$, denoted by $\operatorname{row}(A_t)$. Further we have $\nabla f_t(w) = 2 A_t^T(A_t w - b_t) \in \operatorname{row}(A_t)$. So the loss $f_t$ exhibits strong curvature along the direction of its gradient too. This is the fundamental reason behind the exp-concavity of $f_t$. The min-max barrier $S_t(w)$ is designed such that its gradient is guaranteed to lie in the $\operatorname{row}(A_t)$ (see Lemma 16 in Appendix for a formal statement). So the overall gradient $\nabla \ell_t(w)$ also lies in the $\operatorname{row}(A_t)$. Since the function $f_t$ already exhibits strong curvature along $\operatorname{row}(A_t)$, we conclude that the sum $\ell_t(w) = f_t(w) + G \cdot S_t(w)$ exhibits strong curvature along its gradient $\nabla \ell_t(w)$. This maintains the exp-concavity of the losses $\ell_t$ over $\mathcal{D}_\infty$ (see Lemma 17 in Appendix). Such curvature considerations along with the fact that $S_t(w)$ has to be sufficiently large to facilitate the instantaneous regret bound $f_t(\hat{w}_t) - f_t(u_t) \le \ell_t(w_t) - \ell_t(u_t)$ results in functional form for $S_t(w)$ displayed in Fig.1.

Consequently the fast dynamic regret rates derived in Baby and Wang [2022] becomes directly applicable. The reduction scheme used by Cutkosky and Orabona [2018] for producing proper iterates $\hat{w}_t$ and their accompanying surrogate loss design $\ell_t$ also allows one to upper bound the regret wrt linear regression losses $f_t$ by the regret of the algorithm $\mathcal{A}$ wrt surrogate losses $\ell_t$. However, the surrogate loss $\ell_t$ they construct is not guaranteed to be exp-concave and consequently not amenable to fast dynamic regret rates.

## 4.3 Main Results

We have the following guarantee for ProDR.control:

**Theorem 5.** *Let $u_{1:n} \in \mathcal{D}$ be any comparator sequence. In Fig.1, choose $G$ such that $\sup_{w_1,w_2 \in \mathcal{D}_\infty(\tilde{R}), t \in [n]} \|A_t(w_1 + w_2) - 2b_t\|_1 \leq G$. Let $\alpha$ be as in Assumption 2. Let $L$ be such that $\sup_{w \in \mathcal{D}_\infty(\tilde{R}), j \in [p]} 2\|A_t w - b_t\|_2^2 + 2G^2 \leq L$ for all $t \in [n]$. Choose $\mathcal{A}$ as the algorithm from Baby and Wang [2022] (see Appendix A) with parameters $\gamma = 2G\alpha\tilde{R}\sqrt{d/8L} + \sqrt{2L}$ and $\zeta = \min\{\frac{1}{16G\alpha\tilde{R}\sqrt{d}}, 1/(4\gamma^2)\}$ and decision set $\mathcal{D}_\infty(\tilde{R})$. Under Assumptions 1 and 2, a valid of assignment of $G$ and $L$ are $2p\chi + 2\sigma$ and $6(p\chi + \sigma)^2$ respectively.*

*Then the algorithm ProDR.control yields a dynamic regret rate of*

$$\sum_{t=1}^{n} f_t(\hat{w}_t) - f_t(u_t) = \tilde{O}(d^3 n^{1/3} [\mathcal{TV}(u_{1:n})]^{2/3} \vee 1),$$

*where $(a \vee b) := \max\{a, b\}$.*

**Remark 6.** *In view of Proposition 10 in Baby and Wang [2021], the dynamic regret guarantee in Theorem 5 is optimal modulo dependencies in $d$ and $\log n$. Further the algorithm does not require apriori knowledge of the path length $\mathcal{TV}(u_{1:n})$.*

***Proof sketch for Theorem 5.*** First step is to show that $f_t(\hat{w}_t) \leq \ell_t(w_t)$. This is accomplished by Lipschitzness type arguments. For any $u \in \mathcal{D}$, one observes that $\ell_t(u) = f_t(u)$. So the instantaneous regret of ProDR.control, $f_t(\hat{w}_t) - f_t(u_t)$, is upper bounded by the instantaneous regret, $\ell_t(w_t) - \ell_t(u_t)$ of the surrogate algorithm $\mathcal{A}$. The crucial step is to show the exp-concavity of the losses $\ell_t$ across $\mathcal{D}_\infty(\tilde{R})$. For this, we prove that there is a sub-gradient $\nabla S_t(w)$ that is aligned with $a_{t,j}$ for some $j \in [p]$. This observation followed by few algebraic manipulations (see proof of Lemma 17 in Appendix) allows us to show the exp-concavity of $\ell_t$ over $\mathcal{D}_\infty(\tilde{R})$. Now the overall regret can be controlled if the surrogate algorithm $\mathcal{A}$ provides optimal dynamic regret under exp-concave losses and box decision sets, $\mathcal{D}_\infty(\tilde{R})$. This is accomplished by choosing $\mathcal{A}$ as the algorithm in Baby and Wang [2022] which is also strongly adaptive. $\qquad\square$

Since the surrogate algorithm $\mathcal{A}$ we used in Theorem 5 is strongly adaptive (see for eg. Appendix A), we also have the following performance guarantee in terms of adaptive regret:

**Proposition 7.** *Consider the instantiation of ProDR.control in Theorem 5. Then for any time window $[a,b] \subseteq [n]$ we have that: $\sum_{t=a}^{b} f_t(\hat{w}_t) - \inf_{u \in \mathcal{D}} \sum_{t=a}^{b} f_t(u) = \tilde{O}(d^{1.5} \log n)$.*

**Remark 8.** *Theorem 5 and Proposition 7 together makes the algorithm ProDR.control a good candidate for performing proper online linear regression in non-stationary environments.*

### 4.4 Linear regression with delayed feedback

In this section, we consider a linear regression protocol with feedback delayed by $\tau$ time steps.

- At round $t$, nature reveals a co-variate matrix $A_t \in \mathbb{R}^{p \times d}$.
- Learner plays $z_t \in \mathcal{D} \subset \mathbb{R}^d$.
- Nature reveals the loss $f_{t-\tau+1}(z) = \|A_{t-\tau+1}z - b_{t-\tau+1}\|_2^2$.

This delayed setting can be handled by the framework developed in Joulani et al. [2013]. Although these authors focus on bounding the regret as a function of time horizon $n$, the extension to dynamic regret bounds expressed in terms of both $n$ and $\mathcal{TV}(u_{1:n})$ can be handled straight-forwardly in the analysis. We include the analysis in Appendix B for the sake of completeness. The entire algorithm is as shown in Fig.2.

We have the following regret guarantee for Algorithm ProDR.control.delayed.

**Theorem 9.** *Let $x_t$ be the prediction of the algorithm in Fig. 2 at time $t$. Instantiating each ProDR.control instance by the parameter setting described in Theorem 5. Let $\tau$ be the feedback delay. We have that*

$$\sum_{t=1}^{n} f_t(x_t) - f_t(u_t) = \tilde{O}(d^3 \tau^{2/3} n^{1/3} [\mathcal{TV}(u_{1:n})]^{2/3} \vee \tau).$$

> ProDR.control.delayed: Inputs- delay $\tau > 0$
> - Maintain $\tau$ separate instances of ProDR.control (Fig.1). Enumerate them by $0, 1, \ldots, \tau - 1$.
> - At time $t$:
>     1. Update instance $(t-1) \mod \tau$ with loss $f_{t-\tau}$.
>     2. Predict using instance $(t-1) \mod \tau$.

Figure 2: ProDR.control.delayed: An instance of delayed to non-delayed reduction from Joulani et al. [2013]

*Further for any interval $[a, b] \subseteq [n]$:*

$$\sum_{t=a}^{b} f_t(x_t) - f_t(u) = O(d^{1.5} \tau \log n).$$

## 5 Instantiation for the LQR Problem

In view of Proposition 2, the LQR problem is reduced to a mini-batch linear regression problem with delayed feedback, where the delay is given by $h = O(\log n)$ in Proposition 2. In this section, we provide explicit form of the linear regression losses arising in the LQR problem and instantiate Algorithm ProDR.control.delayed (Fig.2). First we need to define certain quantities:

For a sequence of matrices $(M^{[i]})_{i=1}^{m}$ define `flatten`$((M^{[i]})_{i=1}^{m})$ as follows: Let $M_k^{[i]}$ be the $k^{th}$ column of $M^{[i]}$.

Let's define

$$z^k = \begin{bmatrix} M_1^k \\ \vdots \\ M_{d_x}^k \end{bmatrix} \in \mathbb{R}^{d_u d_x},$$

and

$$\texttt{flatten}((M^{[i]})_{i=1}^{m}) := \begin{bmatrix} z^1 \\ \vdots \\ z^m \end{bmatrix} \in \mathbb{R}^{m d_u d_x}.$$

For a sequence of DAP parameters $M_{1:n}$, let $\mathcal{TV}(M_{1:n}) := \sum_{t=2}^{n} \sum_{i=1}^{m} \|M_t^{[i]} - M_{t-1}^{[i]}\|_1$. We define `deflatten` as the natural inverse operation of `flatten`. We have the following Corollary of Theorem 9 and Proposition 2.

**Corollary 10.** *Assume the notations in Fig.1 and Section 2. Let $\Sigma_\infty = U_\infty^T \Lambda_\infty U_\infty$ be the spectral decomposition of the positive semi definite (PSD) matrix $\Sigma_\infty \in \mathbb{R}^{d_u \times d_u}$. . Let the covariate matrix $A_t := [w_{t-1}^T \ldots w_{t-m}^T] \otimes \Lambda_\infty^{1/2} U_\infty \in \mathbb{R}^{d_u \times m d_u d_x}$, where $\otimes$ denotes the Kronecker product. Let the bias vector $b_t := \Lambda_\infty^{1/2} U_\infty q_{\infty;h}^*(w_{t:t+h})$. Let the delay factor of ProDR.control.delayed (Fig.2) be $\tau = h$ as defined in Proposition 2 and let the decision set given to the ProDR.control instances in Fig.2 be the DAP space defined in Eq.(2). Let $z_t$ be the prediction at round $t$ made by the ProDR.control.delayed algorithm and let $M_t^{alg} := \texttt{deflatten}(z_t)$. At round $t$, we play the control signal $u_t^{alg}(x_t) = \pi_t^{M_t^{alg}}(x_t)$ according to Eq.(1). There exists a choice of input parameters (see Corollary 19 in Appendix B) for the ProDR.control instances in Fig.2 such that*

$$R(M_{1:n}) = \sum_{t=1}^{n} \ell(x_t^{alg}, u_t^{alg}) - \ell(x_t^{M_{1:n}}, u_t^{M_{1:n}})$$

$$= \tilde{O}\left(m^3 d^4 d_x^5 (d_u \wedge d_x)(n^{1/3}[\mathcal{TV}(M_{1:n})]^{2/3} \vee 1)\right),$$

where $M_{1:n}$ is a sequence of DAP policies where each $M_t \in \mathcal{M}$ (eq.(2)). Further the algorithm ProDR.control.delayed also enjoys a strongly adaptive regret guarantee for any interval $[a, b] \subseteq [n]$:

$$\sum_{t=a}^{b} \ell(x_t^{alg}, u_t^{alg}) - \ell(x_t^M, u_t^M) = \tilde{O}((md_u d_x)^{1.5} \log n),$$

for any fixed DAP policy $M \in \mathcal{M}$.

The following theorem provides a nearly matching lower bound.

**Theorem 11.** *There exists an LQR system, a choice of the perturbations $w_t$ and a DAP policy class such that:*

$$\sup_{M_{1:n} \text{ with } \mathcal{TV}(M_{1:n}) \leq C_n} \mathbb{E}[R(M_{1:n})] = \Omega(n^{1/3} C_n^{2/3} \vee 1),$$

*where the expectation is taken wrt randomness in the strategies of the agent and adversary.*

We refer the reader to Appendix B for a proof of the above theorem along with its connections to online non-parametric regression framework of Rakhlin and Sridharan [2014].

**Remark 12.** *The covariate matrix $A_t \in \mathbb{R}^{d_u \times md_u d_x}$ that arises in Corollary 10 is rank deficient whenever $md_x > 1$. In such cases, the linear regression losses $f_t(w)$ as in Fig.1 cannot be strongly convex. So the proper universal dynamic regret minimizing algorithm for strongly convex losses from Baby and Wang [2022] is inapplicable in general except potentially for the particular setting of $m = d_x = 1$. Moreover, in the setting of $m = d_x = 1$ a non-zero strong convexity parameter can exist only if the magnitude of the perturbations $|w_t|$ are bounded away from zero which is restrictive in its scope.*

## 6 Conclusion and Future Work

In this paper, we proposed a new algorithm for minimizing dynamic regret of the non-stationary linear regression problem. We applied this algorithm to obtain a non-stationary LQR controller. The techniques developed in this work can be of independent interest in the broader literature of online learning. We defer the task of deriving similar dynamic regret rates for general strongly convex losses in the LQR problem as a future work.

As mentioned in Section 1, there has been a recent surge of interest in applying tools from online learning to develop non-stochastic controllers. The present work also falls under this umbrella. However, existing literature lacks experimental studies in this vein. It would be a good direction to do a thorough survey of the strengths and limitations of various online learning based controllers when deployed in practice.

## Acknowledgements

The construction of the lower bound in Theorem 11 is due to an early discussion with Daniel Lokshtanov on a related problem. We also thank the anonymous reviewers for their detailed suggestions in improving the paper.

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
