# A Brief overview of results from Baby and Wang [2022]

For the sake of completeness, we dedicate this session for a short discussion about the results of Baby and Wang [2021].

First, we recall the description of Follow-the-Leading-History (FLH) algorithm from [Hazan and Seshadri, 2007].

---

FLH: inputs - Learning rate $\zeta$ and $n$ base learners $E^1, \ldots, E^n$

1. For each $t$, $v_t = (v_t^{(1)}, \ldots, v_t^{(t)})$ is a probability vector in $\mathbb{R}^t$. Initialize $v_1^{(1)} = 1$.

2. In round $t$, set $\forall j \leq t$, $x_t^j \leftarrow E^j(t)$ (the prediction of the $j^{th}$ bas learner at time $t$). Play $x_t = \sum_{j=1}^{t} v_t^{(j)} x_t^{(j)}$.

3. After receiving $f_t$, set $\hat{v}_{t+1}^{(t+1)} = 0$ and perform update for $1 \leq i \leq t$:

$$\hat{v}_{t+1}^{(i)} = \frac{v_t^{(i)} e^{-\zeta f_t(x_t^{(i)})}}{\sum_{j=1}^{t} v_t^{(j)} e^{-\zeta f_t(x_t^{(j)})}}$$

4. Addition step - Set $v_{t+1}^{(t+1)}$ to $1/(t+1)$ and for $i \neq t+1$:

$$v_{t+1}^{(i)} = (1 - (t+1)^{-1})\hat{v}_{t+1}^{(i)}$$

---

Figure 3: FLH algorithm

Next, we describe Online Newton Step (ONS) algorithm from Hazan et al. [2007].

---

ONS: inputs - $\zeta$. Decision set $\mathcal{D}$.

1. At round 1, predict 0.

2. At iteration $t > 1$ predict:

$$w_t \in \underset{x \in \mathcal{D}}{\operatorname{argmin}} \|w_{t-1} - \frac{1}{\beta} A_{t-1}^{-1} \nabla_{t-1} - x\|_{A_{t-1}},$$

where $\nabla_\tau = \nabla f_\tau(x_\tau)$, $A_t = \zeta I_d + \sum_{i=1}^{t} \nabla_i \nabla_i^\top$.

---

Figure 4: ONS algorithm

**Assumption A1:** The loss functions $\ell_t$ are $\alpha$ exp-concave in the box decision set $\mathcal{D} = \{x \in \mathbb{R}^d : \|x\|_\infty \leq B\}$ .ie, $\ell_t(y) \geq \ell_t(x) + \nabla \ell_t(x)^T(y-x) + \frac{\alpha}{2}\left(\nabla \ell_t(x)^T(y-x)\right)^2$ for all $x, y \in \mathcal{D}$.

**Assumption A2:** The loss functions $\ell_t$ satisfy $\|\nabla \ell_t(x)\|_2 \leq G$ and $\|\nabla \ell_t(x)\|_\infty \leq G_\infty$ for all $x \in \mathcal{D}$. Without loss of generality, we let $G \wedge G_\infty \wedge B \geq 1$, where $a \wedge b := \min\{a, b\}$.

We consider the following protocol:

- At time $t \in [n]$ learner predicts $x_t \in \mathbb{R}^d$ with $\|x_t\|_\infty \leq B$.
- Adversary reveals the loss function $\ell_t$.

In view of Assumption A1, following [Hazan et al., 2007], one can define the surrogate losses:

$$f_t(x) = \left(\sqrt{\alpha/2}\nabla \ell_t(x_t)^T(x - x_t) + 1/\sqrt{2\alpha}\right)^2.$$

The surrogate losses satisfy the following property:

$$\sum_{t=1}^{n} \ell_t(x_t) - \ell_t(w_t) \leq \sum_{t=1}^{n} f_t(x_t) - f_t(w_t),$$

where $x_t, w_t \in \mathcal{D}$.

We have the following dynamic regret guarantee from Baby and Wang [2021].

**Theorem 13.** *Suppose Assumptions A1-A2 are satisfied. Define $\gamma := 2GB\sqrt{\alpha d/2} + 1/\sqrt{2\alpha}$. By using the base learner as ONS with parameter $\zeta = \min\left\{\frac{1}{16GB\sqrt{d}}, 1/(4\gamma^2)\right\}$, decision set $\mathcal{D}$, loss at time $t$ to be $f_t$ and choosing learning rate of FLH as $\eta = 1/(2\gamma^2)$, FLH-ONS obeys*

$$\sum_{t=1}^{n} \ell_t(x_t) - \ell_t(w_t) \leq \tilde{O}\left(140d^2(8G^2B^2\alpha d + G^2B^2 + 1/\alpha)(n^{1/3}[\mathcal{TV}(u_{1:n})]^{2/3} \vee 1)\right)\mathbb{I}\{\mathcal{TV}(u_{1:n}) > 1/n\}$$

$$+ \tilde{O}\left(d(8G^2B^2\alpha d + 1/\alpha)\mathbb{I}\{\mathcal{TV}(u_{1:n}) \leq 1/n\}\right),$$

*where $x_t$ is the decision of the algorithm at time $t$ and $\tilde{O}(\cdot)$ hides polynomial factors of $\log n$. $\mathbb{I}\{\cdot\}$ is the boolean indicator function assuming values in $\{0, 1\}$.*

We also have the following strongly adaptive regret guarantee:

**Theorem 14.** *Consider the setting of FLH-ONS in Theorem 13. Then for any interval $[a, b] \subseteq [n]$ we have that*

$$\sum_{t=a}^{b} \ell_t(x_t) - \ell_t(w_t) = O(d^{1.5}\log n).$$

## B    Omitted Proofs

In this section we use the notations defined in Fig.1.

The lemma below shows how the surrogate losses $\ell_t$ can be used to upper bound the regression losses $f_t$.

**Lemma 15.** *Assume the notations in Fig.1. Let $G$ be such that $\sup_{w_1,w_2\in\mathcal{D}_\infty(\tilde{R})}\|A_t(w_1 + w_2) - 2b_t\|_1 \leq G$ for all $t \in [n]$. We have that:*

- *$f_t(\hat{w}_t) \leq \ell_t(w_t)$,*
- *$f_t(u) = \ell_t(u)$ for all $u \in \mathcal{D}$*

*Proof.* For any $w_1, w_2 \in \mathcal{D}_\infty(\tilde{R})$

$$\begin{aligned}
f_t(w_1) - f_t(w_2) &= \|A_tw_1 - b_t\|_2^2 - \|A_tw_2 - b_t\|_2^2 \\
&= (A_t(w_1 + w_2) - 2b_t)^T (A_t(w_1 - w_2)) \\
&\leq \|A_t(w_1 + w_2) - 2b_t\|_1\|A_t(w_1 - w_2)\|_\infty \\
&\leq G \max_{i=1,\ldots,p}|a_{t,i}^T(w_1 - w_2)|,
\end{aligned} \tag{6}$$

for a $G$ such that $\sup_{w_1,w_2\in\mathcal{D}_\infty(\tilde{R})}\|A_t(w_1 + w_2) - 2b_t\|_1 \leq G$ holds true.

In particular we have that:

$$f_t(\hat{w}_t) \leq f_t(w_t) + G \max_{i=1,\ldots,p}|a_{t,i}^T(\hat{w}_t - w_t)| := \ell_t(w_t)$$

For any $u \in \mathcal{D}$, we have that $S_t(u) = 0$. Hence $f_t(u) = \ell_t(u)$.

$\square$

The lemma below establishes certain useful properties of the barrier function $S_t(w)$.

**Lemma 16.** *The function $S_t(w)$ satisfies the following properties:*

1. *$S_t(w) = \max_{i=1,\ldots,p} \min_{x\in\mathcal{D}}|a_{i,t}^T(x - w)|$.*

2. $S_t(w)$ is convex over $\mathbb{R}^d$.

3. Let $i^*$ be such that $S_t(w) = \min_{x \in \mathcal{D}} |a_{i^*,t}^T(x-w)|$. Let $\Pi(w) \in \operatorname{argmin}_{x \in \mathcal{D}} |a_{i^*,t}^T(x-w)|$. Let $g_t \in \partial S_t(w)$, When $a_{i^*,t}^T(\Pi(w)-w) \neq 0$ we have:

$$g_t = \begin{cases} a_{i^*,t}, & \text{if} \quad a_{i^*,t}^\top(\Pi(w)-w) < 0 \\ -a_{i^*,t}, & \text{if} \quad a_{i^*,t}^\top(\Pi(w)-w) > 0. \end{cases}$$

If $a_{i^*,t}^T(\Pi(w)-w) = 0$ then we take $g_t = 0$.

*Proof.* We set out to prove the first statement. Let $\Delta_p$ be the $p$ dimensional simplex. We have that

$$S_t(w) = \min_{x \in \mathcal{D}} \max_{i=1,\ldots,p} |a_{i,t}^T(x-w)|$$

$$=_{(a)} \min_{x \in \mathcal{D}} \max_{v \in \Delta_p} \sum_{i=1}^{p} v_i |a_{i,t}^T(x-w)|$$

$$=_{(b)} \max_{v \in \Delta_p} \min_{x \in \mathcal{D}} \sum_{i=1}^{p} v_i |a_{i,t}^T(x-w)|.$$

For line (a) we observed that for a given $x$ $\max_{v \in \Delta_p} \sum_{i=1}^{p} v_i |a_{i,t}^T(x-w)|$ is attained by putting all the weights of $v$ to an $i^* \in \operatorname{argmax}_{i=1,\ldots,p} |a_{i,t}^T(x-w)|$.

For line (b) we observe that the function $r(x,v) = \sum_{i=1}^{p} v_i |a_{i,t}^T(x-w)|$ is a convex function of $x$ and concave function of $p$. So by applying Sion's minimax theorem we arrive at line (b).

Next we set out to prove that:

$$\max_{v \in \Delta_p} \min_{x \in \mathcal{D}} r(x,v) = \max_{i=1,\ldots,p} \min_{x \in \mathcal{D}} |a_{i,t}^T(x-w)| \tag{7}$$

Let $(x^*, v^*)$ be a solution that attains $\max_{v \in \Delta_p} \min_{x \in \mathcal{D}} r(x,v)$. Further, for the sake of contradiction, let's assume that $v^* \neq e_k$ for any $k \in [p]$. ($e_k$ is the unit vector with 1 at entry $k$). Let the index $j$ be such that $|a_{j,t}^T(x^*-w)| > |a_{i,t}^T(x^*-w)|$ for all $i \in [p] \setminus \{j\}$. Then we can find a solution $e_j$ such that $r(x^*, e_j) > r(x^*, v^*)$. This contradicts the fact that $(x^*, v^*)$ is a valid solution.

In the alternate case let $j$ be an index in $[p]$ such that $|a_{j,t}^T(x^*-w)| \geq |a_{i,t}^T(x^*-w)|$ for all $i \in [p] \setminus \{j\}$. Suppose for all $i \in Q \subseteq [p] \setminus \{j\}$ we have $|a_{j,t}^T(x^*-w)| = |a_{i,t}^T(x^*-w)|$. By earlier arguments, we must have $v^*[k]$ must be equal to zero for all $k \in [p] \setminus (Q \cup \{j\})$. Then putting all the weight to $j$ produces an equally valid solution in the sense that $r(x^*, e_j) = r(x^*, v^*)$

Combining the above two cases, we conclude that there exists maximizers $v^*$ such that $v^* = e_k$ for some $k \in [p]$. This leads to Eq.(7).

Next we prove statement 2. For any given $i$ we have that $|a_{i,t}^T(x-w)|$ is a convex function of both $x$ and $w$. Hence the point-wise maximum $\max_{i=1,\ldots,p} |a_{i,t}^T(x-w)|$ is also convex in both $x$ and $w$. Since partial minimisation preserves convexity, we have that $\min_{x \in \mathcal{D}} \max_{i=1,\ldots,p} |a_{i,t}^T(x-w)|$ remains convex in $w \in \mathbb{R}^d$.

Next we prove statement 3. We know that sub-gradient set of point-wise maximum of convex functions is the convex hull of sub-gradients of the active functions. Applying this result along with the sub-gradient characterization of the function $\min_{x \in \mathcal{D}} |a_{i,t}^T(x-w)|$ in Lemma 18 leads to the third statement.

$\square$

The next lemma establishes the exp-concavity of the surrogate losses $\ell_t$ over the decision domain of the surrogate algorithm $\mathcal{A}$.

**Lemma 17.** *Assume the notations in Fig.1. Let $L$ be such that $\sup_{w \in \mathcal{D}_\infty(\tilde{R}), j \in [p]} 2\|A_t w - b_t\|_2^2 + 2G^2 \leq L$ for all $t \in [n]$. Then the losses $\ell_t$ are exp-concave over $\mathcal{D}_\infty(\tilde{R})$ with parameter $1/4L$.*

*Proof.* Observe that $\nabla f_t(w) = 2A_t^T(A_t w - b_t)$ and $\nabla^2 f_t(w) = 2A_t^T A_t$.

We have that for any $w_1, w_2 \in \mathbb{R}^d$

$$f_t(w_2) = f_t(w_1) + \langle \nabla f_t(w_1), w_2 - w_1 \rangle + \frac{1}{2}\|w_2 - w_1\|^2_{2A_t^T A_t}. \tag{8}$$

Due to the convexity of $S_t(w)$ over $\mathbb{R}^d$ from Lemma 16, we have that

$$S_t(w_2) \geq S(w_1) + \langle \nabla S_t(w_1), w_2 - w_1 \rangle. \tag{9}$$

Combining Eq.(8) and (9) we have that

$$\ell_t(w_2) \geq \ell_t(w_1) + \langle \nabla \ell_t(w_1), w_2 - w_1 \rangle + \frac{1}{2}\|w_2 - w_1\|^2_{2A_t^T A_t}$$

Observe that $\nabla \ell_t(w_1) = 2A_t^T(A_t w_t - b_t) + GhA_t^T e_j$, for some $h \in \{-1, 0, 1\}$ and $j \in [p]$ due to Lemma 16. Now, let's focus on points $w_1, w_2 \in \mathcal{D}_\infty(\tilde{R})$. We have

$$\nabla \ell_t(w_1) \nabla \ell_t(w_1)^T = 4A_t^T(A_t w_1 - b_t + Ghe_j)(A_t w_1 - b_t + Ghe_j)^T A_t$$
$$\preccurlyeq 4L A_t^T A_t,$$

$L$ is such that:

$$\sup_{w \in \mathcal{D}_\infty(\tilde{R}), j \in [p]} \|(A_t w - b_t + Ghe_j)\|_2^2 \leq L.$$

Hence for all $w_1, w_2 \in \mathcal{D}_\infty(\tilde{R})$, we have the relation

$$\ell_t(w_2) \geq \ell_t(w_1) + \langle \nabla \ell_t(w_1), w_2 - w_1 \rangle + \frac{1}{4L}\|w_2 - w_1\|^2_{\nabla \ell_t(w_1) \nabla \ell_t(w_1)^T}.$$

Thus the losses $\ell_t$ remains exp-concave over $\mathcal{D}_\infty(\tilde{R})$ with parameter $1/4L$.

$\square$

We are now ready to prove Theorem 5.

**Theorem 5.** *Let $u_{1:n} \in \mathcal{D}$ be any comparator sequence. In Fig.1, choose $G$ such that $\sup_{w_1, w_2 \in \mathcal{D}_\infty(\tilde{R}), t \in [n]} \|A_t(w_1 + w_2) - 2b_t\|_1 \leq G$. Let $\alpha$ be as in Assumption 2. Let $L$ be such that $\sup_{w \in \mathcal{D}_\infty(\tilde{R}), j \in [p]} 2\|A_t w - b_t\|_2^2 + 2G^2 \leq L$ for all $t \in [n]$. Choose $\mathcal{A}$ as the algorithm from Baby and Wang [2022] (see Appendix A) with parameters $\gamma = 2G\alpha \tilde{R}\sqrt{d/8L} + \sqrt{2L}$ and $\zeta = \min\{\frac{1}{16G\alpha \tilde{R}\sqrt{d}}, 1/(4\gamma^2)\}$ and decision set $\mathcal{D}_\infty(\tilde{R})$. Under Assumptions 1 and 2, a valid of assignment of $G$ and $L$ are $2p\chi + 2\sigma$ and $6(p\chi + \sigma)^2$ respectively.*

*Then the algorithm ProDR.control yields a dynamic regret rate of*

$$\sum_{t=1}^{n} f_t(\hat{w}_t) - f_t(u_t) = \tilde{O}(d^3 n^{1/3}[\mathcal{TV}(u_{1:n})]^{2/3} \vee 1),$$

*where $(a \vee b) := \max\{a, b\}$.*

*Proof.* From Eq.(6) we have that for any $w_1, w_2 \in \mathcal{D}_\infty(\tilde{R})$

$$f_t(w_1) - f_t(w_2) \leq G\alpha \|w_1 - w_2\|_2,$$

for a $G$ such that $\sup_{w_1, w_2 \in \mathcal{D}_\infty(\tilde{R})} \|A_t(w_1 + w_2) - 2b_t\|_1 \leq G$ holds true.

From Lemma 16 we have for any subgradient $\|\nabla S_t(w)\|_2 \le \alpha$ (where $\alpha$ is as in Assumption 1). Thus the losses $\ell_t$ are $2G\alpha$-Lipschitz in L2 norm over $\mathcal{D}_\infty(\tilde{R})$. Now combining Lemma 17 and Theorem 10 in Baby and Wang [2022] (or see Appendix A) we have that

$$\sum_{t=1}^{n} \ell_t(w_t) - \ell_t(u_t) = \tilde{O}\left((d^3 G^2 \alpha^2 \tilde{R}^2/L + d^2 G^2 \alpha^2 \tilde{R}^2 + d^2 L)(n^{1/3}[\mathcal{TV}(u_{1:n})]^{2/3} \vee 1)\right)$$

$$= \tilde{O}(d^3 n^{1/3}[\mathcal{TV}(u_{1:n})]^{2/3} \vee 1).$$

Applying Lemma 15 now concludes the proof. $\qquad\square$

**Lemma 18.** *Let $f(x) = min_{u \in \mathcal{D}}|a^T(u-x)|$ for a compact and convex set $\mathcal{D}$. Let $0 \in \mathcal{D}$. Then:*

- *$f(x)$ is convex.*
- *Let $s \in argmin_{u \in \mathcal{D}}|a^T(u-x)|$. Then,*

$$\nabla f(x) = \begin{cases} -a & a^T(s-x) > 0 \\ a & a^T(s-x) < 0 \\ 0 & o.w \end{cases}$$

*Proof.* First we argue the convexity of $f$. Observe that

$$f(x) = \min_{u \in \mathcal{D}}|a^T(u-x)|$$

$$= \min_{u \in \mathcal{D}}\|u-x\|_{aa^T}.$$

The norm $\|u-x\|_{aa^T}$ is convex in both $u$ and $x$ across $\mathbb{R}^d$. So we have that $f(x)$ which is obtained by partial minimization of a convex function across a convex domain remains convex over $\mathbb{R}^d$. It follows that for any $x, y \in \mathbb{R}^d$,

$$f(y) \ge f(x) + \nabla f(x)^T(y-x). \tag{10}$$

Next, we proceed to show the Lipschitzness of $f$. Let $w \in argmin_{u \in \mathcal{D}}|a^T(u-x)|$. We have

$$f(y) - f(x) = \min_{u \in \mathcal{D}}|a^T(u-y)| - \min_{u \in \mathcal{D}}|a^T(u-x)|$$

$$\le |a^T(w-x)| - |a^T(w-y)|$$

$$\le |a^T(x-y)|$$

$$\le \|a\|_2\|x-y\|_2. \tag{11}$$

Since $\|a\|_2 \le \kappa$, we conclude that the function $f$ is $\kappa$ Lipschitz.

We argue that $\nabla f(x) = \lambda a$ for some scalar $\lambda$. Let $b$ be a such that $a^T b = 0$. Let $z = y + \sigma b$. Notice that by the definition of $f$, we have that $f(y) = f(z)$. So,

$$f(z) = f(y)$$

$$\ge f(x) + \nabla f(x)^T(z-x)$$

$$= f(x) + \nabla f(x)^T(y-x) + \sigma \nabla f(x)^T b.$$

The above inequality must hold for any $\sigma \in \mathbb{R}$. Note that both $f(y)$ and $f(x)$ is bounded for any two points in $x, y \in \mathbb{R}^d$. Further, $\nabla f(x)^T(y-x)$ is also bounded due to the Lipschitzness of $f$. So if $\nabla f(x)^T b$ is not zero, we can choose a $\sigma$ such that inequality is violated, leading to a contradiction in the convexity of $f$ across $\mathbb{R}^d$.

So $\nabla f(x)^T b = 0$. This implies that $\nabla f(x) = \lambda(x)a$ for some scalar $\lambda(x)$ and for any $x \in \mathbb{R}^d$.

Next, we argue that $\lambda(x) \in [-1, 1]$. Combining Eq.(10) and (11) we have

$$|a^T(x-y)| \geq \nabla f(x)^T(y-x),$$

for all $x, y \in \mathbb{R}^d$. So taking $y = 0$ followed by $y = 2x$ leads to

$$|a^T x| \geq \pm\lambda(x)a^T x.$$

Suppose $x$ is chosen such that $a^T x \neq 0$. Then the above inequality implies that $\lambda(x) \in [-1, 1]$.

Let $w \in argmin_{u \in \mathcal{D}} |a^T(u-x)|$. Let $s = (x+w)/2$. We have that

$$f(s) \geq f(x) + \lambda(x)a^T(s-x). \tag{12}$$

Moroever,

$$\begin{aligned} f(s) &\leq |a^T(w-s)| \\ &= \frac{1}{2}|a^T(x-w)| \\ &= f(x) - |a^T(x-s)|. \end{aligned} \tag{13}$$

Combining Eq.(14) and (15), we obtain

$$-|a^T(s-x)| \geq \lambda(x)a^T(s-x).$$

Recall that when $a^T x \neq 0$, $\lambda(x) \in [-1, 1]$.

So we conclude that if $a^T x \neq 0$ and $a^T(s-x) > 0$, then $\lambda(x) \leq -1$. This implies that $\lambda(x) = -1$ as $\lambda(x) \in [-1, 1]$ holds true.

Similarly if $a^T x \neq 0$ and $a^T(s-x) < 0$, then $\lambda(x) \geq 1$. This implies that $\lambda(x) = 1$ as $\lambda(x) \in [-1, 1]$ holds true.

Now if $a^T x \neq 0$ and $a^T(s-x) = 0$, we can choose $\lambda(x) = 0$ as $f(z) \geq f(x) + \lambda(x)a^T(z-x) = 0$ holds true for any $z$.

If $a^T x = 0$, $0 \in argmin_{u \in \mathcal{D}} |a^T(u-x)|$ as $0 \in \mathcal{D}$ is assumed to be true. So by using the previous line of arguments we conclude that $\lambda(x) = 0$. $\square$

**Theorem 9.** *Let $x_t$ be the prediction of the algorithm in Fig. 2 at time $t$. Instantiating each ProDR.control instance by the parameter setting described in Theorem 5. Let $\tau$ be the feedback delay. We have that*

$$\sum_{t=1}^{n} f_t(x_t) - f_t(u_t) = \tilde{O}(d^3 \tau^{2/3} n^{1/3} [\mathcal{TV}(u_{1:n})]^{2/3} \vee \tau).$$

*Further for any interval $[a, b] \subseteq [n]$:*

$$\sum_{t=a}^{b} f_t(x_t) - f_t(u) = O(d^{1.5} \tau \log n).$$

*Proof.* By following the arguments in Joulani et al. [2013], we have that

$$\sum_{t=1}^{n} f_t(x_t) - f_t(u_t) = \sum_{i=1}^{\tau} \sum_{k=1}^{\lfloor 1 + \frac{n-i}{\tau} \rfloor} f_t(x_{i+(k-1)\tau}) - f_t(u_{i+(k-1)\tau}).$$

The second summation in the above expression is the dynamic regret of instance $i$ wrt comparator sequence $\{u_{i+(k-1)\tau}\}$ with $k$ ranging from 1 to $\lfloor 1 + \frac{n-i}{\tau} \rfloor$. Now by triangle inequality we have that

$$\sum_{k=2}^{\lfloor 1+\frac{n-i}{\tau} \rfloor} \|u_{i+(k-1)\tau} - i + (k-2)\tau\|_1 \leq \sum_{t=2}^{n} \|u_t - u_{t-1}\|_1 = \mathcal{TV}(u_{1:n}).$$

Thus by Theorem 5 we have

$$\sum_{t=1}^{n} f_t(x_t) - f_t(u_t) \leq \sum_{i=1}^{\tau} \tilde{O}(d^3 (n/\tau)^{1/3} \vee 1)$$
$$\leq \tilde{O}(d^3 \tau^{2/3} n^{1/3} [\mathcal{TV}(u_{1:n})]^{2/3} \vee \tau).$$

$\square$

Next, we provide the version of Corollary 10 indicating the closed form expression for all the algorithm parameters.

**Corollary 19.** *Let $\Sigma_\infty = U_\infty^T \Lambda_\infty U_\infty$ be the spectral decomposition of the positive semi definite (PSD) matrix $\Sigma_\infty \in \mathbb{R}^{d_u \times d_u}$. Assume the notations in Fig.1. Let the covariate matrix $A_t := [w_{t-1}^T \ldots w_{t-m}^T] \otimes \Lambda_\infty^{1/2} U_\infty$, where $\otimes$ denotes the Kronecker product. Let the bias vector $b_t := \Lambda_\infty^{1/2} U_\infty q_{\infty;h}^*(w_{t:t+h})$. For a sequence of DAP parameters $M_{1:n}$, let $\mathcal{TV}(M_{1:n}) := \sum_{t=2}^{n} \sum_{i=1}^{m} \|M_t^{[i]} - M_{t-1}^{[i]}\|_1$. For a sequence of matrices $(M^{[i]})_{i=1}^m$ define $\mathtt{flatten}((M^{[i]})_{i=1}^m)$ as follows: Let $M_k^{[i]}$ be the $k^{th}$ column of $M^{[i]}$.*

*Let's define*

$$z^k = \begin{bmatrix} M_1^k \\ \vdots \\ M_{d_x}^k \end{bmatrix} \in \mathbb{R}^{d_u d_x},$$

*and*

$$\mathtt{flatten}((M^{[i]})_{i=1}^m) := \begin{bmatrix} z^1 \\ \vdots \\ z^m \end{bmatrix} \in \mathbb{R}^{m d_u d_x}.$$

*Let the decision set given to the ProDR.control (Fig.1) algorithm be the DAP space defined in Eq.(2). Let $G = 2m d_u d_x R \gamma \sqrt{d_x \wedge d_u} \|\Lambda^{1/2} U_\infty\|_1 + 2 \frac{\|\Lambda^{-1/2} U_\infty B^T\|_2 \|P_\infty\|_2 \sqrt{d_u}}{1-\gamma}$. Let the delay factor of ProDR.control.delayed (Fig.2) be $\tau = h$ as defined in Proposition 2. Choose $\alpha = \sqrt{m \|\Sigma_\infty\|_{op}}$ and $L = 4G^2$. Let $\tilde{R}$ in Theorem 5 be chosen as $\tilde{R} = R\gamma \sqrt{d_u \wedge d_x}$. Let $z_t$ be the prediction at round $t$ made by the ProDR.control.delayed algorithm. Let $M_t^{alg} := \mathtt{deflatten}(z_t)$, where $\mathtt{deflatten}$ is the natural inverse operation of $\mathtt{flatten}$ defined above. Let $\pi := (M_1, \ldots, M_n)$ define a sequence of DAP policies. For a sequence of matrices $M$, define $\|M\|_1 := \sum_{i=1}^{m} \|M^{[i]}\|_1$. By playing a control $u_t^{alg}(x_t) = \pi_t^{M_t^{alg}}(x_t)$ according to Eq.(1), we have that*

$$R_n(M_{1:n}) = \sum_{t=1}^{n} \ell(x_t^{alg}, u_t^{alg}) - \ell(x_t^{M_{1:n}}, u_t^{M_{1:n}}) = \tilde{O}\left( m^3 d^4 d_x^5 (d_u \wedge d_x)(n^{1/3} [\mathcal{TV}(M_{1:n})]^{2/3} \vee 1) \right),$$

*where $M_{1:n}$ is a sequence of DAP policies where each $M_t \in \mathcal{M}$ (eq.(2)). Further the algorithm ProDR.control.delayed also enjoys a strongly adaptive regret guarantee for any interval $[a, b] \subseteq [n]$:*

$$\sum_{t=a}^{b} \ell(x_t^{alg}, u_t^{alg}) - \ell(x_t^M, u_t^M) = \tilde{O}((m d_u d_x)^{1.5} \log n),$$

*for any fixed DAP policy $M \in \mathcal{M}$.*

*Proof.* Define

$$X_t = [w_{t-1}^T \dots w_{t-m}^T] \otimes I_{d_u},$$

where $I_{d_u} \in \mathbb{R}^{d_u \times d_u}$ is the identity matrix and $\otimes$ denotes the Kronecker product. Clearly $X_t \in \mathbb{R}^{d_u \times m d_u d_x}$.

With these definitions, it is easy to verify that

$$q^M(w_{t-1}) = X_t z.$$

Now we return back to losses $\hat{A}_t$ mentioned in Proposition 2. Let $\Sigma_\infty = U_\infty^T \Lambda_\infty U_\infty$ be the spectral decomposition of the positive semi definite (PSD) matrix $\Sigma_\infty \in \mathbb{R}^{d_u \times d_u}$. We have that

$$\hat{A}_t(M; w_{t+h}) = \|\Lambda_\infty^{1/2} U_\infty q^M(w_{t-1}) - \Lambda_\infty^{1/2} U_\infty q_{\infty;h}^*(w_{t:t+h})\|_2^2$$
$$= \|\Lambda_\infty^{1/2} U_\infty X_t z - \Lambda_\infty^{1/2} U_\infty q_{\infty;h}^*(w_{t:t+h})\|_2^2.$$

Define

$$A_t := \Lambda_\infty^{1/2} U_\infty X_t$$
$$= [w_{t-1}^T \dots w_{t-m}^T] \otimes \Lambda_\infty^{1/2} U_\infty$$

Next, we proceed to compute a box that encloses all DAP policies of interest. We have for each $i \in [m]$,

$$\|z^i\|_\infty^2 \le \|z^i\|_2^2$$
$$= \|M^{[i]}\|_F^2$$
$$\le (d_u \wedge d_x)\|M^{[i]}\|_{op}^2$$
$$\le (d_u \wedge d_x)R^2\gamma^2,$$

where the last line is due to the DAP policy set that we are interested in.

Thus the box $\mathcal{D}_\infty(R\gamma\sqrt{d_u \wedge d_x}) := \mathcal{D}_\infty(\tilde{R})$ encapsulates the DAP policy space that we are interested in.

We need to compute the parameters in Theorem 5. First, let's focus on computing $G$. We have for any $z_1, z_2 \in$

$$\|A_t(z_1 + z_2) - 2b_t\|_1 \le 2\|A_t\|_1 m d_u d_x \tilde{R} + 2\|b_t\|_1, \tag{14}$$

where $b_t = \Lambda_\infty^{1/2} U_\infty q_{\infty;h}^*(w_{t:t+h})$.

We have

$$\|A_t\|_1 = \max_{i=1,\dots,m} \|w_{t-i}\|_\infty \|\Lambda^{1/2} U_\infty\|_1$$
$$\le \|\Lambda^{1/2} U_\infty\|_1, \tag{15}$$

as the disturbances obey $\|w_t\|_2 \le 1$.

We have

$$\|b_t\|_2 \le \sum_{i=t}^{t+h} \|\Lambda^{-1/2} U_\infty B^T (A_{cl,\infty})^{i-t} P_\infty w_i\|_2$$
$$\le \sum_{i=t}^{t+h} \|\Lambda^{-1/2} U_\infty B^T\|_2 \|(A_{cl,\infty})^{i-t}\|_2 \|P_\infty\|_2 \|w_i\|_2$$
$$\overset{(a)}{\le} \|\Lambda^{-1/2} U_\infty B^T\|_2 \|P_\infty\|_2 \sum_{i=1}^{h} \gamma^{i-1}$$
$$\le \|\Lambda^{-1/2} U_\infty B^T\|_2 \|P_\infty\|_2 \cdot \frac{1}{1-\gamma},$$

where in line (a) we used the strong stability criterion and the fact that $\|w_t\|_2 \leq 1$. Thus we have

$$\|b_t\|_1 \leq \sqrt{d_u}\|b_t\|_2$$
$$\leq \frac{\|\Lambda^{-1/2} U_\infty B^T\|_2 \|P_\infty\|_2 \sqrt{d_u}}{1-\gamma}. \tag{16}$$

Putting together Eq.(14).(15) and (16) we arrive at

$$\|A_t(z_1 + z_2) - 2b_t\|_1 \leq 2m d_u d_x R\gamma\sqrt{d_x \wedge d_u}\|\Lambda^{1/2} U_\infty\|_1 + 2\frac{\|\Lambda^{-1/2} U_\infty B^T\|_2 \|P_\infty\|_2 \sqrt{d_u}}{1-\gamma}$$
$$:= G \tag{17}$$

Next we proceed to calculate $\alpha$ in Theorem 5. Denote by $U_j$ the $j^{th}$ column of the matrix $U_\infty$. The squared norm of the $i^{th}$ row of the covariate matrix $A_t$ is given by

$$\sum_{k=1}^{m}\|w_{t-k}\|_2^2 \sum_{j=1}^{d_u}\lambda_j u_j^2[i] \leq \|\Sigma_\infty\|_{op}\sum_{k=1}^{m}\sum_{j=1}^{d_u}u_j^2[i]$$
$$= m\|\Sigma_\infty\|_{op},$$

where we used the fact the matrix $U_\infty$ is orthogonal. Thus we choose

$$\alpha = \sqrt{m\|\Sigma_\infty\|_{op}}.$$

By similar arguments used to reach Eq.(17), we choose

$$L = 4G^2$$

For a sequence of policies $M_1, \ldots, M_n$, observe that $\sum_{t=2}^{n}\|\texttt{flatten}(M_t) - \texttt{flatten}(M_{t-1})\|_1 \leq d_x \sum_{t=2}^{n}\|M_t - M_{t-1}\|_1$. The last relation expresses the dynamic regret incurred by ProDR.control.delayed in terms of total variation of $\texttt{flatten}(M_t)$ to be bounded by total variation of the matrices themselves.

Putting all the constants together and applying Theorem 9 and Theorem 5 yields the Corollary.

$\square$

**Theorem 11.** *There exists an LQR system, a choice of the perturbations $w_t$ and a DAP policy class such that:*

$$\sup_{M_{1:n} \text{ with } \mathcal{TV}(M_{1:n}) \leq C_n} \mathbb{E}[R(M_{1:n})] = \Omega(n^{1/3} C_n^{2/3} \vee 1),$$

*where the expectation is taken wrt randomness in the strategies of the agent and adversary.*

*Proof.* Consider a system with matrices $A = 0 \in \mathbb{R}^{2 \times 2}$, $B = \begin{bmatrix} -1 & 0 \\ 0 & -1 \end{bmatrix}$, $R_x = \begin{bmatrix} 1 & 0 \\ 0 & 0 \end{bmatrix}$ and $R_u = 0 \in \mathbb{R}^{2 \times 2}$. In this setting $K_\infty = 0$ as per Eq.(4). We consider DAP polices (see Definition 1) with $m = 1$. Let the starting state be $x_1 = 0 \in \mathbb{R}^{2 \times 2}$.

Let $y_t = \pm 1$ with probability half each. Let $w_t = [y_t, 1]^T$. For a policy that chooses a control signal $u_t$ at time $t$, its next state is given by $x_{t+1} = w_t - u_t$ and $\ell_{t+1}(x_{t+1}, u_{t+1}) = (u_t[1] - y_t)^2$. Hence for any algorithm, the loss is given by:

$$\sum_{t=1}^{n}\ell_t(x_t, u_t) = \sum_{t=1}^{n-1}(u_t^{\text{alg}}[1] - y_t)^2. \tag{18}$$

Divide the time horizon into bins of width $W$. Let the number of bins be $M := n/W$. We assume that $n/W$ is an integer for simplicity. Let the $i^{th}$ be denoted by $[s_i, e_i]$ for $i \in [M]$. Define

$$a_i := \frac{1}{W}\sum_{t=s_i}^{e_i}y_t.$$

We will uniformly use the same DAP policy within a bin $i$ as the comparator. This policy will be parameterized by the matrix $M_i := \begin{bmatrix} 0 & -a_i \\ 0 & 0 \end{bmatrix}$

By Hoeffding's inequality and a union bound across all $M$ bins, we arrive at

$$a_i \in \left[ -\sqrt{\frac{\log(nM/\delta)}{2W}}, \sqrt{\frac{\log(nM/\delta)}{2W}} \right],$$

with probability at-least $1 - \delta$. We will call this high probability event as $\mathcal{E}$. Due to symmetry we have that $P(y_t = 1|\mathcal{E}) = 1/2$. So under the event $\mathcal{E}$, the Bayes optimal online prediction of any algorithm as per Eq.(18) will be to set $u = [0,0]^T$. So within a bin we have that

$$\sum_{t=1}^{n} E[\ell_t(x_t, u_t)|\mathcal{E}] \geq W.$$

Now we need to upper bound the cumulative loss of the comparator within a bin. Since the policy within a bin is parameterized by $M_i$, we have that $u_t = -M_t w_{t-1} = [a_i, 0]^T$ for all $t \in [s_i, e_i]$.

So we have:

$$
\begin{aligned}
E[(y_t - u_t)^2|\mathcal{E}] &= \frac{E[(y_t - u_t)^2] - E[(y_t - u_t)^2|\mathcal{E}^c]P(\mathcal{E}^c)}{P(\mathcal{E})} \\
&\leq \frac{E[(y_t - u_t)^2]}{1 - \delta},
\end{aligned}
$$

where $\mathcal{E}^c$ denotes complement of event $\mathcal{E}$.

By bias variance decomposition, we have that

$$E[(y_t - u_t)^2] = 1 - 1/W.$$

So the overall regret is lower bounded by

$$
\begin{aligned}
\sum_{i=1}^{M} \sum_{t=s_i}^{e_i} E[(y_t - u_t^{\text{alg}}[1])^2|\mathcal{E}] - E[(y_t - a_i)^2|\mathcal{E}] &\geq \sum_{i=1}^{M} W(1 - \frac{1}{1-\delta}) + \frac{1}{1-\delta} \\
&\geq M/(1-\delta) - W\delta/(1-\delta) \\
&\geq M/2, \quad (19)
\end{aligned}
$$

where the last line is obtained by setting $\delta = 1/n^2$

Under the event $\mathcal{E}$ with $\delta = 1/n^2$, the total variation (TV) of the sequence $a_{1:n}$ is given by:

$$\mathcal{TV}(a_{1:n}) \leq \frac{n\sqrt{2\log(n^4)}}{W^{3/2}}.$$

Now setting $W = \frac{n^{2/3}(8\log n)^{1/3}}{C_n^{2/3}}$ we obtain $\mathcal{TV}(a_{1:n}) \leq C_n$ with probability at-least $1 - 1/n^2$.

Continuing from Eq.(19), we obtain that

$$
\begin{aligned}
E[R_n|\mathcal{E}] := \sum_{i=1}^{M} \sum_{t=s_i}^{e_i} E[(y_t - u_t^{\text{alg}}[1])^2|\mathcal{E}] - E[(y_t - a_i)^2|\mathcal{E}] \\
\geq \frac{n^{1/3}C_n^{2/3}}{2(8\log n)^{1/3}}, \quad (20)
\end{aligned}
$$

where the event $\mathcal{E}$ occurs with probability at-least $1 - 1/n^2$.

When $C_n \leq 1/\sqrt{n}$, the static regret bound of $\Omega(\log n)$ (see Theorem 11.9 in Cesa-Bianchi and Lugosi [2006]). This completes the proof of the theorem.

$\square$

**Connections to online non-parametric regression framework of Rakhlin and Sridharan [2014].**
In the work of Rakhlin and Sridharan [2014], they study the following online regression framework (simplified here without affecting the information-theoretic rates):

- At each round $t$, learner plays a decision $x_t \in \mathbb{R}$.
- Nature reveals a label $y_t$ such that $|y_t| \leq 1$.
- Learner suffers loss $(y_t - x_t)^2$.

One is interested in finding the min-max rate of regret against a non-parametric sequence class. We define the space of total variation (TV) bounded sequences as:

$$\mathcal{TV}(C_n) := \{\theta_{1:n} | \mathcal{TV}(\theta_{1:n}) \leq C_n\}.$$

Translated into the setup of Rakhlin and Sridharan [2014], one can aim to control the regret against $\mathcal{TV}(C_n)$ which is:

$$R_n := \sum_{t=1}^{n} (y_t - x_t)^2 - \inf_{\theta_{1:n} \in \mathcal{TV}(C_n)} \sum_{t=1}^{n} (y_t - \theta_t)^2. \tag{21}$$

The TV class is known to be sandwiched between two Besov spaces having the same minimax rate (see for eg. [DeVore and Lorentz, 1993]). So the results of Rakhlin and Sridharan [2014] based on characterizing the sequential Rademacher complexity of the Besov class leads to $O(n^{1/3})$ as the minimax rate of $R_n$ wrt $n$. The rate wrt $C_n$ was not provided in their work. However, we remark that they establish an $O(n^{1/3})$ upper bound also via non-constructive arguments.

In contrast, the lower bound we provided in the proof of Theorem 11 is for $\sum_{t=1}^{n} E[(y_t - u_t^{\text{alg}}[1])^2 - (y_t - a_t)^2 | \mathcal{E}]$ (Eq.(20)) where $\mathcal{TV}(a_{1:n}) \leq C_n$ under the high probability event $\mathcal{E}$ trivially lower bounds $R_n$ in Eq.(21) with high probability.