# OpenReview forum: "Optimal Dynamic Regret in LQR Control"
_NeurIPS.cc/2022/Conference — NeurIPS 2022 Accept_

### Official Review · Reviewer_Eo6g · 2022-07-02

**Rating:** 6
**Confidence:** 4
**Soundness:** 3 good
**Presentation:** 3 good
**Contribution:** 2 fair

**Summary:**

The paper proposes an algorithm which attains an $\tilde{O}(n^{1/3} \mathcal{TV}(M_{1:n})$ dynamic regret in the nonstochastic LQR setting, along with a matching lower bound. The analysis builds upon the reduction of nonstochastic quadratic-loss LQR to online linear regression with delayed feedback of Foster and Simchowitz (2020) and the recent algorithm of Baby and Wang (2022) attaining optimal dynamic regret (and $O(\log n)$ strongly adaptive regret) for exp-concave losses under certain assumptions. Hence the main theorem of the paper translates the Baby and Wang (2022) guarantees to the setting of \textit{proper} (i.e. ensuring decisions are within a general convex set) online linear regression with delayed feedback and then instantiates it in the LQR setting.

**Questions:**

Suggestions/minor remarks:
- On line 294: "static" -> "adaptive";
- line 355: "arise"->"arises"
- Some more text to help the flow of Appendix B (quite terse right now)
- misplaced equation in lines 657-660
- Improve flow of proof of lemma 18 (e.g. bullet points for properties, or just more careful writing + organization)
- line 694: paragraph ends with "We fo" -> fill in the rest of the sentence
- Shorten statement of Corollary 19 to improve readability + maybe some light work on improving flow of proof
- the below.

**Limitations:**

I recommend more discussion on finding the correct input parameters for Corollary 10 to hold (line 333) [addressed in rebuttal, I recommend this to be incorporated in camera-ready], as well as discussion of the limits of deploying online learning based controllers in practice (both in general and the particular one proposed by the authors) [I recommend this to be incorporated in discussion section of camera-ready].

**Strengths And Weaknesses:**

Strengths:
- The (main) paper is written very clearly;
- It is placed very naturally within the literature;
- The conversion of Baby and Wang (2022) to an algorithm that works on general convex constraint sets is a nice contribution.

Weaknesses:
- Somewhat limited originality and impact;
- Lack of experimental settings make it hard to gauge if this algorithm would eventually lead to better controllers in practice.

---

> ### Author Response · Authors · 2022-08-01
> **Respone to Reviewer Eo6g**
>
> We thank the reviewer for appreciating our contributions and finding that the paper is clearly written. We promise to incorporate all stylistic suggestions.
>
> **Input parameters in Corollary 10:** Unfortunately due to space constraints, we couldn't explain the exact parameter settings at Corollary 10. Instead, we wrote it in Corollary 19 in supplementary. We will try to accommodate it in the main text.
>
> **About deploying online learning based controllers in practice:** Combining ideas from online learning and control theory has been gaining a lot of momentum recently [eg. 3,6,7]. However, we are unfamiliar with papers showcasing experimental results in this vein. It would be a good direction to do a thorough survey of the strengths and limitations of various online learning based controllers when deployed in practice.
>
> **References**
>
> [3] Dylan J. Foster and Max Simchowitz, Logarithmic Regret for Adversarial Online Control, ICML 2020
>
> [6] Elad Hazan, Sham M. Kakade and Karan Singh, The Nonstochastic Control Problem, ALT 2020
>
> [7] Max Simchowitz, Making Non-Stochastic Control (Almost) as Easy as Stochastic, NuerIPS 2020

---

> > ### Comment · Reviewer_Eo6g · 2022-08-07
> > **Thank you!**
> >
> > Thank you for your reply! Since camera-ready provides an additional page, I do recommend trying to fit in the parameter settings in main text for camera-ready, or at least a hyperlink to corollary 19. I would recommend adding some variation of the second comment "it would be a good direction to do a thorough survey of the strengths and limitations of various online learning based controllers when deployed in practice" in your conclusion. I completely agree that unfortunately most of the works on online learning for control lack experimental validation (so this is not specific to this work), but I do believe it makes sense to straightforwardly address this limitation/direction for future work in the text. As a final note, I maintain my score (and slightly edited my review with the above recommendations).

---

### Official Review · Reviewer_95P1 · 2022-07-10

**Rating:** 6
**Confidence:** 3
**Soundness:** 3 good
**Presentation:** 3 good
**Contribution:** 3 good

**Summary:**

This work studies LQR control problem w.r.t. the dynamic regret measure. The authors propose a ProDR algorithm that achieves an optimal $n^{1/3}$ regret, improving the existing best $n^{1/2}$ regret result. The main idea of their algorithm is to reduce the original LQR problem to an online linear regression problem and adopt a ProDR online linear regression algorithm with an $n^{1/3}$ regret. They further prove an $n^{1/3}$ regret lower bound to suggest the optimality of their algorithm.

**Questions:**

- Can you highlight the main technical difficulties you have faced in applying the algorithms in Baby and Wang (2021) to the LQR setting? I briefly checked Baby and Wang (2021) (https://arxiv.org/pdf/2201.08905.pdf) and it seems that they have already discussed some reduction from the box constraints to the general convex decision sets in their Sec. 4.2. Thus, I am not sure whether your comment from line 228 to 229 are still applicable.

**Limitations:**

The authors have discussed the limitations of their algorithms.

**Strengths And Weaknesses:**

Strengths:

+ The presentation is clear.
+ The proof is technically sound.
+ The relation between the proposed algorithm and existing baseline algorithms is well discussed.

Weaknesses:

- The importance of the derived result seems not important enough since it heavily depends on an existing algorithm from Baby and Wang (2021). I am not sure if the transferring framework (from LQR to online linear regression) is also a standard process or not. If so, it may further weaken the technical contribution of this work.

---

> ### Author Response · Authors · 2022-08-01
> **Response to Reviewer 95P1**
>
> We thank the reviewer for going through our proofs and appreciating the presentation of the paper.
>
> We believe the reviewer is referring to the work of [4] in their review. Please correct us if we got it wrong.
>
> **LQR to online linear regression:** We remark that the regret in Eq.3 is dynamic *policy* regret. The states visited by the reference policy is counterfactual and is different from that of the learner's trajectory which we observe. This is very different from the standard online convex optimization (OCO) framework where the state of both the learner and adversary are same. So bounding the policy regret seems qualitatively harder than bounding the regret in an OCO setting. Nevertheless, for the LQR problem, the fact that there exists a reduction [3] from the problem of controlling policy regret to the problem of controlling the standard OCO regret is remarkable. To the best of our knowledge [3] is the only work that goes through this route of reduction from LQR to online linear regression. Though they consider a static regret setting, our work backs up that their reduction is just as significant in non-stationary environments also by designing a new algorithm for non-stationary proper online linear regression to control dynamic regret.
>
>
> **Technical difficulties in applying the results of [4]:** As the reviewer noted, [4] provides algorithms to control dynamic regret with general convex decision sets. For the sake of clarity we briefly summarise results in [4]:
>
> * [4] provides a proper online algorithm to optimally control dynamic regret under strongly convex losses and general convex decision sets.
>
> * [4] provides a proper online algorithm to optimally control dynamic regret under exp-concave losses and *box* ($L_\infty$ constrained) decision sets.
>
> However, the main caveat here is that the reduction in [4] to support general convex decision sets only works for *strongly convex* losses as opposed to *exp-concave* losses. Linear regression losses are indeed not strongly convex. Linear regression losses are only exp-concave. In fact [4] applies the reduction scheme in [5] unmodified  to support general convex decision sets. However, as noted in Section 4.2 this route of applying the reduction of [5] unmodified destroys the exp-concavity of the surrogate losses $\ell_t(w)$ (in Line 4 of Fig.1) that we pass to the algorithm $\mathcal A$. Doing so prevents us from using the algorithm of [4] as $\mathcal A$ that can work with exp-concave losses and box decision sets to control the dynamic regret of ProDR.control. To bypass the issue, we design new surrogate losses based on a min-max barrier which guarantees that the surrogates $\ell_t(w)$ are exp-concave. This makes it possible to choose $\mathcal A$ as the algorithm of [4] that works with box decision sets and exp-concave losses and convert it to an algorithm that support general decision sets under linear regression. We remark that the design of such surrogate losses which preserve exp-concavity is non-trivial in hindsight. Please also see Remark 12 for a comment on the limitations of applying the results of [4] for the LQR problem.
>
> **References**
>
> [3] Dylan J. Foster and Max Simchowitz, Logarithmic Regret for Adversarial Online Control, ICML 2020
>
> [4] Dheeraj Baby and Yu-Xiang Wang Optimal Dynamic Regret in Proper Online Learning with Strongly Convex Losses and Beyond, AISTATS 2022
>
> [5] Ashok Cutkosky and Francesco Orabona, Black-Box Reductions for Parameter-free Online Learning in Banach Spaces, COLT 2018

---

### Official Review · Reviewer_63zh · 2022-07-12

**Rating:** 6
**Confidence:** 3
**Soundness:** 4 excellent
**Presentation:** 3 good
**Contribution:** 3 good

**Summary:**

The paper studies the dynamic regret of online linear quadratic control. It considers single-trajectory LQ control with known, stationary system and cost matrices, and with adversarial disturbances. The paper addresses the problem by studying dynamic regret minimization against disturbance action policies. It first presents a proper online learning algorithm that achieves dynamic regret optimal in the number of steps and path variation, for online minibatch linear regression. The paper proves that this algorithm is also strongly adaptive, in that it achieves logarithmic static regret for any time window. By combining disturbance action policies with the proper online learning procedure, the paper shows a dynamic regret bound that is also optimal in the dependence on the number of steps and path variation. A main technical contribution is the procedure to convert an improper online learning algorithm to a proper one, while maintaining the exp-concavity of the loss function. A nontrivial lower bound is also presented to show the optimal dependencies.

**Questions:**

In the abstract, citing the source of the best known rate is recommended.

In some places, the papers uses the notation $\tilde{O}(\log n)$. Here, $\tilde{O}(\cdot)$ does not hide logarithmic dependence on $n$, which had better be noted.

Below Proposition 2, in Lines 133 to 137, the math formulas can be better displayed.

In the definition of exp-concavity (Definition 3), it is not clear how it is related to the name of exp-concavity. Unfamiliar readers would appreciate some more explanation.

In Section 4.2, readers read the algorithm first but do not know what $\mathcal{A}$ is until reading two more paragraphs. I recommend explaining $\mathcal{A}$ earlier.

In Line 326, "notations" -> "notation".

In the Conclusion section, the paper mentions "general strongly convex losses in the LQR problem". If I did not get it wrong, this paper considers positive semidefinite $Q$, $R$, so the cost is not necessarily strongly convex. Hence, strongly convex costs are not strictly more general than quadratic costs. The word "general" can be removed to reduce the ambiguity. And in this case, the problem is no longer a LQR problem, but a linear control problem.

**Limitations:**

Limitations are adequately addressed.

**Strengths And Weaknesses:**

The paper has technical significance. It proves a sharp dynamic regret bound in terms of the number of steps and the path variation, as well as a matching lower bound. The results are built upon earlier results about the reduction from online LQ control to online linear regression, optimal improper online learning with exp-concave losses, and the conversion from improper learners to proper ones, while having new, interesting components, like the new improper to proper learning conversion and lower bounding strategy.

The paper contains a thorough discussion of the literature and is well placed in it. The presentation is in general clear. Please see the Questions section for some suggestions for improvement.

On the weakness side, the dependence on the dimension and system parameters are unlikely to be optimal. Empirical support for the theoretical claims is lacking, which could be informative of the constants hidden in the bounds.

---

> ### Author Response · Authors · 2022-08-01
> **Response to Reviewer 63zh**
>
> We thank the reviewer for appreciating our technical contributions. We promise to incorporate the writing suggestions you mentioned.
>
> **Source of best known rate:** The rate mentioned in the abstract is due to the algorithm in [2]. We will cite it in the abstract. The main reason for the slow rate is that [2] only considers general convex losses without exploiting the curvature of the losses.
>
> **Exp-concavity:** A loss function $f(x)$ is $\alpha$ exp-concave, whenever $e^{-\alpha f(x)}$ is concave. The characterization in Definition 3 is due to [1] (Lemma 2 there).
>
> **Description of $\mathcal A$:** We will provide a detailed description of $\mathcal A$ earlier itself as it will improve clarity. We remark that ProDR.control is truly a black-box reduction and *any* algorithm $\mathcal A$ for online linear regression can be applied here. If we choose $\mathcal A$ to be as in Theorem 5, then it provides optimal performance guarantees.
>
> **References**
>
> [1] Elad Hazan, Amit Agarwal and Satyen Kale,  Logarithmic Regret Algorithms for Online Convex Optimization, Kluwer Academic Publishers, 2007; http://www.satyenkale.com/papers/HKKA2006.pdf
>
> [2] Peng Zhao, Yu-Xiang Wang and Zhi-Hua Zhou, Non-stationary Online Learning with Memory and Non-stochastic Control, AISTATS 2022

---

> > ### Comment · Reviewer_63zh · 2022-08-08
> > **Response to authors**
> >
> > Thank you for addressing my questions. After reading the author response, I maintain my judgement in favor of acceptance.

---

### Meta-Review · Area_Chair_SAYC · 2022-08-27

**Recommendation:** Accept
**Confidence:** Less certain

**Metareview:**

This paper considers the problem of online linear-quadratic control with adversarial noise. Previous work aims to derive regret bounds relative to the best linear controller in hindsight, whereas this work considers *dynamic regret*, where the algorithm ensures regret against all sequences of policies simultaneously, with the regret depending on the variation in the sequence. The main result is to provide an algorithm with policy regret $n^{1/3}\cdot\mathrm{Variation}^{2/3}$, which improves upon the previous state of the art.

The reviewers found this paper to be clear and well-written, and found the paper technically interesting (the main challenge is to extend the dynamic regret guarantees of Baby and Wang (2022) guarantees to the setting of \emph{proper} online learning, so that the reduction from online control to online regression of Foster and Simchowitz (2020) can be applied. For the final revision, the authors are encouraged to incorporate the reviewers' feedback to improve the presentation.

**Award:**

No

---

### Decision · Program_Chairs · 2022-09-14

Accept